# CDC-42 Interactions with Par Proteins Are Critical for Proper Patterning in Polarization

**DOI:** 10.3390/cells9092036

**Published:** 2020-09-05

**Authors:** Sungrim Seirin-Lee, Eamonn A. Gaffney, Adriana T. Dawes

**Affiliations:** 1Department of Mathematics and Department of Mathematical and Life Sciences, Graduate School of Integrated Sciences for Life, Hiroshima University, Higashi-Hiroshima 739-8530, Japan; 2Mathematical Institute, University of Oxford, Oxford OX2 6GG, UK; gaffney@maths.ox.ac.uk; 3Department of Mathematics, The Ohio State University, Columbus, OH 43210, USA; 4Department of Molecular Genetics, The Ohio State University, Columbus, OH 43210, USA

**Keywords:** intracellular polarization, partial differential equations, sensitivity analysis

## Abstract

Many cells rearrange proteins and other components into spatially distinct domains in a process called polarization. This asymmetric patterning is required for a number of biological processes including asymmetric division, cell migration, and embryonic development. Proteins involved in polarization are highly conserved and include members of the Par and Rho protein families. Despite the importance of these proteins in polarization, it is not yet known how they interact and regulate each other to produce the protein localization patterns associated with polarization. In this study, we develop and analyse a biologically based mathematical model of polarization that incorporates interactions between Par and Rho proteins that are consistent with experimental observations of CDC-42. Using minimal network and eFAST sensitivity analyses, we demonstrate that CDC-42 is predicted to reinforce maintenance of anterior PAR protein polarity which in turn feedbacks to maintain CDC-42 polarization, as well as supporting posterior PAR protein polarization maintenance. The mechanisms for polarity maintenance identified by these methods are not sufficient for the generation of polarization in the absence of cortical flow. Additional inhibitory interactions mediated by the posterior Par proteins are predicted to play a role in the generation of Par protein polarity. More generally, these results provide new insights into the role of CDC-42 in polarization and the mutual regulation of key polarity determinants, in addition to providing a foundation for further investigations.

## 1. Introduction

Intracellular polarization, whereby a cell establishes a pattern and specifies a spatial axis by segregating proteins and other factors to distinct domains, is a fundamental and ubiquitous process. Polarization is implicated in a wide variety of biological phenomena including asymmetric cell division, cell migration, wound healing, and embryonic development [1,2,3]. Aberrant polarization is also thought to play a role in disease progression: a hallmark of the epithelial to mesenchymal transition (EMT) in malignant cells is the acquisition of a polarized migratory phenotype [4,5]. The same key polarity determinants, including the Par and Rho protein families, are required for forming the pattern associated with polarization in virtually all cell types and organisms [3]. Despite the importance of polarization and the highly conserved nature of the proteins involved, the mechanisms and signalling networks regulating this patterning process are not completely understood.

Two main protein families have been intensively studied for their role in polarization: Rho proteins and Par proteins (Figure 1). Rho proteins, also referred to as Rho GTPases, are monomeric G proteins that interconvert between an active GTP-bound state, and an inactive GDP-bound state. The active form of the protein can associate with the membrane at the periphery of the cell, while the inactive form is found diffusing in the cytoplasm [6]. Three members of this family, Rac, Rho and Cdc-42, have been extensively studied for their role in cell migration [6,7,8,9]. Polarized migrating cells establish front and back domains, with Cdc-42 and Rac segregating to the front, and Rho associating with the rear of the cell. Par proteins were first identified in the early embryos of the nematode worm *C. elegans* for their role in properly patterning the embryo prior to first division [3,10,11]. The asymmetric localization of the Par proteins specifies the anterior/posterior axis of the developing embryo, and is required for the asymmetric first division. PAR-3, PAR-6 and the atypical protein kinase aPKC bind to the membrane on the periphery of the cell and specify the anterior half of the cell, while PAR-1 and PAR-2 bind to the membrane on the posterior half. Loss of polarity in the first cell cycle is lethal to the embryo [12]. Recent experimental observations suggest that Par and Rho proteins rely on mutual interaction and feedback to produce the pattern of proteins associated with polarization: T cells require both Cdc-42 and Par proteins for polarization [13], and *C. elegans* embryos require CDC-42 for proper patterning of PAR-2 and PAR-6 [14,15]. Despite their involvement in polarization, the dynamics of Par and Rho proteins have been studied largely independently from each other.

While polarization is observed in many different systems, we focus on protein patterning in the early *C. elegans* embryo [16] (Figure 2). The establishment phase, which is initiated by fertilization, moves symmetrically distributed Par proteins, including PAR-3, PAR-6 and aPKC, to the anterior half of the cell. The cleared area in the posterior half is then available for binding by the posterior Par proteins, including PAR-1 and PAR-2. Once the polarized domains are established, the cell transitions into the maintenance phase, where it maintains the asymmetric protein pattern as the cell prepares for first division. While the protein transport associated with polarization is important for establishment of polarization [17], we focus on the role of biochemical interactions in the generation and maintenance of polarized domains, independent of advective flow. Members of the Rho protein family, notably CDC-42, are thought to be important for polarization in the early embryo, but how they interact with and regulate Par proteins is not clear.

In this paper, we develop a continuum model of Par and Rho protein dynamics in the generation and maintenance of polarization in the early *C. elegans* embryo from the rational simplification for the underlying interactions among CDC-42 and the PAR proteins that have been uncovered by numerous experimental studies. This model also utilizes the large ratio of diffusive transport scales between the cell cytosol and membrane and a simple representation of the cell geometry. We demonstrate that the resulting partial differential equation model is consistent with observations of CDC-42, highlighting the requirement for CDC-42 during the maintenance of polarization in the early *C. elegans* [18,19]. To elucidate the detailed mechanism of interactions capable of reproducing experimental observations, we perform sensitivity analysis and minimal network analysis to identify and characterize predictions for key cross talk and mutual regulation interactions among CDC-42 and the PAR proteins in controlling the generation and maintenance of cellular polarization. Due to the conserved nature of both the process and proteins involved in polarization, the insights gained in this study apply broadly to other biological systems beyond *C. elegans*.

## 2. The Mathematical Model

### 2.1. Network of Par Protein and Cdc-42 Interactions

To investigate the role potential role of interactions between Par and Rho protein family members in the generation and maintenance of distinct spatial domains, we construct a network of interactions that is consistent with experimental data. Justification for each arrow in Figure 1A is as follows:(i)PAR-6/PKC-3/PAR-3 promotes dissociation of PAR-1 from the membrane to the cytosol [10,16,20].(ii)PKC-3/PAR-3 promotes dissociation of PAR-2 from the membrane to the cytosol [20,21,22].(iii)PAR-2 promotes dissociation of PAR-6, PAR-3, and CDC-42 from the membrane to the cytosol [22].(iv)PAR-1 promotes dissociation of PAR-3 from the membrane to the cytosol [23].(v)PAR-3 associates with PAR-6, allowing PAR-6 to be maintained in the membrane [22].(vi)CDC-42 associates with PAR-6/PKC-3, allowing PAR-6/PKC-3 to be maintained in the membrane [18,20,22,23,24].(vii)PAR-6 associates with CDC-42, allowing CDC-42 to be maintained in the membrane [18,22,24,25].(viii)PAR-2 associates with PAR-1, preventing PAR-1 dissociation from the membrane by PAR-6/PKC-3/PAR-3 [10,26].

Figure 1A shows a full schematic diagram and the network of interactions corresponding to the descriptions (i)–(viii). Some of these interactions may not be direct, but are mediated by other proteins. For instance, enhanced cortical dissociation of the anterior Par proteins by PAR-2 is likely mediated by PAR-1 [27]. In addition, the role of CDC-42 and its interactions with the Par proteins are not entirely clear, and some observations are not accounted for in our model, such as the appearance of CDC-42 on the cortex even in the absence of PAR-6 [14]. Nevertheless, experimental observations of all interactions listed above have been reported in the literature.

To determine the fundamental network of interactions based on the above experimental evidence, we separated the interaction network in Figure 1A into three subnetworks, Figure 1B(b1,b2,b4). In Figure 1B(b1), we consider the interactions between [PAR-6, PAR-2 and PAR-1] and [PKC-3, PAR-2 and PAR-1]. In this subnetwork, the inhibition of PAR-1 by PAR-6 and inhibition of PAR-6 by PAR-2 (dotted green line) is equivalent to PAR-2 activation of PAR-1 (solid green line). Since PAR-6 and PKC-3 only dissociate PAR-1 and PAR-2 when PAR-6 and PKC-3 are bound together in a complex, we can reduce this subnetwork to a mutual inhibition between [PAR-6 and PKC-3] and [PAR-1 and PAR-2] (Figure 1B(b1)). In a similar manner, the subnetwork containing [PAR-3, PAR-2 and PAR-1] can be reduced to a network of mutual inhibition between [PAR-3] and [PAR-1 and PAR-2] (Figure 1B(b2)). We further simplify these subnetworks by grouping the Par proteins according to their localization: anterior Par proteins (aPAR) and posterior Par proteins (pPAR) (Figure 1B(b3)). Finally, we combine the mutual Par protein inhibition subnetwork with the [CDC-42, PAR-6/PKC-3 and PAR-2] subnetwork (Figure 1B(b4)) to arrive at the fundamental interaction network of [aPAR, pPAR, and CDC-42] (Figure 1C).

### 2.2. Model Equations for Apar, Ppar and Cdc-42 Network

We define the cytosol by $\Omega \subset R^{3}$ and the membrane by ∂Ω(≡Γ) so that Ω∪∂Ω represents *C. elegans* single cell embryo (Figure 2A). We represent the concentrations of anterior proteins (aPARs, i.e., PAR-6, PAR-3, and PKC-3) in the membrane and the cytosol by $\left[A_{m}\left(x,t\right)\right]$ and $\left[A_{c}\left(x,t\right)\right]$, respectively, and the concentrations of posterior proteins (pPARs, i.e., PAR-1 and PAR-2) in the membrane and the cytosol by $\left[P_{m}\left(x,t\right)\right]$ and $\left[P_{c}\left(x,t\right)\right]$, respectively, and the concentrations of CDC-42 in the membrane and the cytosol by $\left[C_{m}\left(x,t\right)\right]$ and $\left[C_{c}\left(x,t\right)\right]$, respectively, where $x\in R^{3}$ and t∈[0,∞). The dynamics of the network shown in Figure 1C can then be described as follows:(1)$$\begin{array}{cccc} \; & \frac{\partial [A_{m}]}{\partial t}=D_{m}^{A}\nabla_{\Gamma }^{2}\left[A_{m}\right]+\left\{\gamma_{a}+F_{\mathrm{on}}^{A}\left(\left[C_{m}\right]\right)\right\}\left[A_{c}\right]-\left\{\alpha_{a}+F_{\mathrm{off}}^{A}\left(\left[P_{m}\right]\right)\right\}\left[A_{m}\right] & \mathrm{on}\; & x\in \partial \Omega ,\\ \; & \frac{\partial [A_{c}]}{\partial t}=D_{c}^{A}\nabla^{2}\left[A_{c}\right] & \mathrm{on}\; & x\in \Omega ,\\ \; & D_{c}^{A}\frac{\partial [A_{c}]}{\partial n}=-\left\{\gamma_{a}+F_{\mathrm{on}}^{A}\left(\left[C_{m}\right]\right)\right\}\left[A_{c}\right]+\left\{\alpha_{a}+F_{\mathrm{off}}^{A}\left(\left[P_{m}\right]\right)\right\}\left[A_{m}\right] & \mathrm{on}\; & x\in \partial \Omega ,\\ \; & \frac{\partial [P_{m}]}{\partial t}=D_{m}^{P}\nabla_{\Gamma }^{2}\left[P_{m}\right]+\gamma_{p}\left[P_{c}\right]-\left\{\alpha_{p}+F_{\mathrm{off}}^{P}\left(\left[A_{m}\right]\right)\right\}\left[P_{m}\right] & \mathrm{on}\; & x\in \partial \Omega ,\\ \; & \frac{\partial [P_{c}]}{\partial t}=D_{c}^{P}\nabla^{2}\left[P_{m}\right] & \mathrm{on}\; & x\in \Omega ,\\ \; & D_{c}^{P}\frac{\partial [P_{c}]}{\partial n}=-\gamma_{p}\left[P_{c}\right]+\left\{\alpha_{a}+F_{\mathrm{off}}^{P}\left(\left[A_{m}\right]\right)\right\}\left[P_{m}\right] & \mathrm{on}\; & x\in \partial \Omega ,\\ \; & \frac{\partial [C_{m}]}{\partial t}=D_{m}^{C}\nabla_{\Gamma }^{2}\left[C_{m}\right]+\gamma_{c}\left[C_{c}\right]-\left\{\alpha_{c}+F_{\mathrm{off}}^{C}\left(\left[P_{m}\right],\left[A_{m}\right]\right)\right\}\left[C_{m}\right] & \mathrm{on}\; & x\in \partial \Omega ,\\ \; & \frac{\partial [C_{c}]}{\partial t}=D_{c}^{C}\nabla^{2}\left[C_{c}\right] & \mathrm{on}\; & x\in \Omega ,\\ \; & D_{c}^{C}\frac{\partial [C_{c}]}{\partial n}=-\gamma_{c}\left[C_{c}\right]+\left\{\alpha_{c}+F_{\mathrm{off}}^{C}\left(\left[P_{m}\right],\left[A_{m}\right]\right)\right\}\left[C_{m}\right] & \mathrm{on}\; & x\in \partial \Omega , \end{array}$$
where $D_{m}^{A}$, $D_{m}^{P}$ and $D_{m}^{C}$ are the diffusion coefficients of aPARs, pPARs and CDC-42 in the membrane, respectively, and $D_{c}^{A}$, $D_{c}^{P}$ and $D_{c}^{C}$ are the diffusion coefficients of aPARs, pPAR and CDC-42 in the cytosol, respectively. n is the inner normal vector on ∂Ω. We assume that $F_{\mathrm{on}}^{A},F_{\mathrm{off}}^{A}$ and $F_{\mathrm{off}}^{P}$ are increasing functions of CDC-42 concentration, pPAR concentration, and aPAR concentration, respectively. We also assume that $F_{\mathrm{off}}^{C}$ is an increasing function of pPAR concentration but a decreasing function of aPAR concentration. For the purposes of this study, we choose the following functional forms [22,28];
(2)$$\begin{array}{cc} \; & F_{\mathrm{on}}^{A}\left(\left[C_{m}\right]\right)=\frac{q_{3}^{a}{\left[C_{m}\right]}^{2}}{q_{1}^{a}+q_{2}^{a}{\left[C_{m}\right]}^{2}},\;F_{\mathrm{off}}^{A}\left(\left[P_{m}\right]\right)=\frac{k_{3}^{a}{\left[P_{m}\right]}^{2}}{k_{1}^{a}+k_{2}^{a}{\left[P_{m}\right]}^{2}},\\ \; & F_{\mathrm{off}}^{P}\left(\left[A_{m}\right]\right)=\frac{k_{3}^{p}{\left[A_{m}\right]}^{2}}{k_{1}^{p}+k_{2}^{p}{\left[A_{m}\right]}^{2}},\;F_{\mathrm{off}}^{C}\left(\left[P_{m}\right],\left[A_{m}\right]\right)=\frac{k_{3}^{c}{\left[P_{m}\right]}^{2}}{k_{1}^{c}+k_{2}^{c}{\left[P_{m}\right]}^{2}}+\frac{q_{3}^{c}}{q_{1}^{c}+q_{2}^{c}{\left[A_{m}\right]}^{2}}. \end{array}$$
Rate constants of the form $k_{i}^{j}$ denote interactions that include pPAR while rate constants of the form $q_{i}^{j}$ denote interactions strictly between aPAR and CDC-42.

Parameter values corresponding to the final simplified model are summarized in Table A1. Note that some parameters and variables are redefined during this simplification process. The first step in this simplification exploits differences in diffusion dynamics. We require $D_{c}^{A}\gg D_{m}^{A}$, $D_{c}^{P}\gg D_{m}^{P}$ and $D_{c}^{C}\gg D_{m}^{C}$ since diffusion in the cytosol is much faster than diffusion in the membrane [29,30]. Furthermore, this fast cytosolic diffusion, if sufficiently large, results in homogeneous spatial distribution of cytoplasmic proteins over the time scale of interest (Figure A1) and allows us to reduce the model (Equation 1) to a shadow system. By taking $D_{c}^{A},D_{c}^{P},D_{c}^{C}\to \infty$, the leading order approximations for cytosolic protein concentrations quickly approach a homogeneous steady states such that
$$\begin{array}{c} A_{c}\left(x,t\right)\to \frac{1}{\left|\Omega \right|}\int_{\Omega }\left[A_{c}\right]\left(x,t\right)dx,\;P_{c}\left(x,t\right)\to \frac{1}{\left|\Omega \right|}\int_{\Omega }\left[P_{c}\right]\left(x,t\right)dx,\;C_{c}\left(x,t\right)\to \frac{1}{\left|\Omega \right|}\int_{\Omega }\left[C_{c}\right]\left(x,t\right)dx. \end{array}$$
The following conservation relations hold:$$\begin{array}{c} \frac{d}{dt}\left(\int_{\Omega }\left[A_{c}\right]\left(x,t\right)dx+\int_{\Gamma }\left[A_{m}\right]\left(x,t\right)dx\right)=0,\\ \frac{d}{dt}\left(\int_{\Omega }\left[P_{c}\right]\left(x,t\right)dx+\int_{\Gamma }\left[P_{m}\right]\left(x,t\right)dx\right)=0,\\ \frac{d}{dt}\left(\int_{\Omega }\left[C_{c}\right]\left(x,t\right)dx+\int_{\Gamma }\left[C_{m}\right]\left(x,t\right)dx\right)=0, \end{array}$$
and the total mass of the model (Equation 1) is conserved. We define the conserved total concentrations as follows:$$\begin{array}{cc} \; & A_{tot}\equiv \int_{\Omega }\left[A_{c}\right]\left(x,t\right)dx+\int_{\Gamma }\left[A_{m}\right]\left(x,t\right)dx,\;P_{tot}\equiv \int_{\Omega }\left[P_{c}\right]\left(x,t\right)dx+\int_{\Gamma }\left[P_{m}\right]\left(x,t\right)dx,\\ \; & C_{tot}\equiv \int_{\Omega }\left[C_{c}\right]\left(x,t\right)dx+\int_{\Gamma }\left[C_{m}\right]\left(x,t\right)dx, \end{array}$$
giving us
$$\begin{array}{cc} \; & A_{c}\left(x,t\right)\approx \frac{1}{\left|\Omega \right|}\left(A_{tot}-\int_{\Gamma }\left[A_{m}\right]\left(x,t\right)dx\right),\;P_{c}\left(x,t\right)\approx \frac{1}{\left|\Omega \right|}\left(P_{tot}-\int_{\Gamma }\left[P_{m}\right]\left(x,t\right)dx\right),\\ \; & C_{c}\left(x,t\right)\approx \frac{1}{\left|\Omega \right|}\left(C_{tot}-\int_{\Gamma }\left[C_{m}\right]\left(x,t\right)dx\right), \end{array}$$
at leading order in an asymptotic approximation based on the cytosolic diffusion dominating all other possible diffusive scales in the model. Thus, working to this level of approximation, model (Equation 1) is reduced to the surface model:(3)$$\begin{array}{cccc} & \frac{\partial [A_{m}]}{\partial t}=D_{m}^{A}\nabla_{\Gamma }^{2}\left[A_{m}\right]+\left\{\gamma_{a}+F_{\mathrm{on}}^{A}\left(\left[C_{m}\right]\right)\right\}\left(\frac{A_{tot}}{\left|\Omega \right|}-\frac{1}{\left|\Omega \right|}\int_{\Gamma }\left[A_{m}\right]dx\right)-\left\{\alpha_{a}+F_{\mathrm{off}}^{A}\left(\left[P_{m}\right]\right)\right\}\left[A_{m}\right] & \mathrm{on}\; & x\in \Gamma ,\\ & \frac{\partial [P_{m}]}{\partial t}=D_{m}^{P}\nabla_{\Gamma }^{2}\left[P_{m}\right]+\gamma_{p}\left(\frac{P_{tot}}{\left|\Omega \right|}-\frac{1}{\left|\Omega \right|}\int_{\Gamma }\left[P_{m}\right]dx\right)-\left\{\alpha_{p}+F_{\mathrm{off}}^{P}\left(\left[A_{m}\right]\right)\right\}\left[P_{m}\right] & \mathrm{on}\; & x\in \Gamma ,\\ & \frac{\partial [C_{m}]}{\partial t}=D_{m}^{C}\nabla_{\Gamma }^{2}\left[C_{m}\right]+\gamma_{c}\left(\frac{C_{tot}}{\left|\Omega \right|}-\frac{1}{\left|\Omega \right|}\int_{\Gamma }\left[C_{m}\right]dx\right)-\left\{\alpha_{c}+F_{\mathrm{off}}^{C}\left(\left[P_{m}\right],\left[A_{m}\right]\right)\right\}\left[C_{m}\right] & \mathrm{on}\; & x\in \Gamma . \end{array}$$

We define the model geometry as the surface of a solid of revolution found by rotating the arc of the membrane about the AP-axis (Figure 2A). Since we are only interested in the dynamics on the membrane, we seek to reduce model (Equation 3) to a one dimensional model on x∈[0,L] where *x* is the arclength along the perimeter of the projected membrane, depicted in Figure 2A, x=0 corresponds to the leftmost point on the membrane and *L* is the cell perimeter. This reduction to a 1D domain also ensures the minimal network and eFAST analysis can be accomplished in a reasonable amount of time. Periodic boundary conditions are imposed at x=0,L (Figure 2A). We further approximate the geometry effectively by a cylinder of radius H≪L with asymptotically short caps and we define the tilde variables and parameters via
$${\tilde{A}}_{m}=\pi HA_{m},\;\;{\tilde{P}}_{m}=\pi HP_{m},\;\;{\tilde{C}}_{m}=\pi HC_{m},\;\;{\tilde{\gamma }}_{a}/\gamma_{a}={\tilde{\gamma }}_{p}/\gamma_{p}={\tilde{\gamma }}_{c}/\gamma_{c}={\tilde{q}}_{3}^{a}/q_{3}^{a}=L\pi H/\left|\Omega \right|.$$

As detailed in Section A.1, we rewrite Equation (Equation 3) in terms of these new parameters, together with the assumption of axisymmetry and geometrical approximations that are stated and justified in Section A.1. On dropping tildes, we have
(4)$$\begin{array}{cc} \; & \frac{\partial [A_{m}]}{\partial t}=D_{m}^{A}\frac{\partial^{2}}{\partial x^{2}}\left[A_{m}\right]+\left\{\gamma_{a}+F_{\mathrm{on}}^{A}\left(\left[C_{m}\right]\right)\right\}\left(\frac{A_{tot}}{L}-\frac{1}{L}\int_{0}^{L}\left[A_{m}\right]dx\right)-\left\{\alpha_{a}+F_{\mathrm{off}}^{A}\left(\left[P_{m}\right]\right)\right\}\left[A_{m}\right],\\ \; & \frac{\partial [P_{m}]}{\partial t}=D_{m}^{P}\frac{\partial^{2}}{\partial x^{2}}\left[P_{m}\right]+\gamma_{p}\left(\frac{P_{tot}}{L}-\frac{1}{L}\int_{0}^{L}\left[P_{m}\right]dx\right)-\left\{\alpha_{p}+F_{\mathrm{off}}^{P}\left(\left[A_{m}\right]\right)\right\}\left[P_{m}\right],\\ \; & \frac{\partial [C_{m}]}{\partial t}=D_{m}^{C}\frac{\partial^{2}}{\partial x^{2}}\left[C_{m}\right]+\gamma_{c}\left(\frac{C_{tot}}{L}-\frac{1}{L}\int_{0}^{L}\left[C_{m}\right]dx\right)-\left\{\alpha_{c}+F_{\mathrm{off}}^{C}\left(\left[P_{m}\right],\left[A_{m}\right]\right)\right\}\left[C_{m}\right]. \end{array}$$

The parameter values in Table A1 are with respect to this final model (Equation (Equation 4)). The model was simulated by custom software written in C. The PDEs were solved via an implicit numerical scheme using standard finite-difference methods. Source code used to generate the results in this paper are available upon request.

### 2.3. Parameter Values

Before symmetry breaking in the *C. elegans* embryo, aPAR is spatially homogeneously distributed on the membrane and pPAR is spatially homogeneously distributed in the cytosol. Gotta et al [19] experimentally demonstrated that loss of CDC-42 by RNAi results in a loss of polarity, with low PAR-6 and high PAR-2 levels on the membrane. Thus, we choose representative kinetic parameters such that aPar, pPar and CDC-42 establish distinct spatial domains under wild type conditions, but fail to polarize when CDC-42 is absent (Figure 2B). For sensitivity analysis, we restrict our parameters as shown in Figure 2C, corresponding to parameter values that are consistent with both wild type and *cdc-42(RNAi)* experimental observations. The fact that such a large range of parameter values produces the appropriate model behaviours provides further validation of the model’s predicted importance for CDC-42 in polarization. Parameter names and definitions for the final model, Equation (Equation 4), are summarized in Table A1, together with the non-dimensionalization used in the numerical investigation of the model.

### 2.4. Initial Conditions

We use two sets of initial conditions to evaluate either maintenance (Section 3.1.1) or generation (Section 3.2.1) of polarization. For simulating maintenance of polarization, we specify polarized initial conditions with aPAR and CDC-42 high in the anterior (left and right) region of the domain and low in the posterior (middle) region, and the reverse profile for pPAR (high in the posterior, low in the anterior). See Figure 2 for a schematic of the model geometry and the anterior/posterior regions. The high and low values are derived from the stationary long time asymptotic solution of Equation (Equation 4) (Figure 2B).

For simulating generation of polarization, we specify initial conditions with a small spatial perturbation:(5)$$\begin{array}{c} \left[P_{m}\right]\left(x,0\right)=\epsilon_{p}\delta \left(x-L/2\right),\;\left[A_{m}\right]\left(x,0\right)=\frac{A_{tot}}{L}\left(1+\epsilon \phi_{a}\left(x\right)\right),\;\left[C_{m}\right]\left(x,0\right)=C_{0}\left(1+\epsilon \phi_{c}\left(x\right)\right), \end{array}$$
where x∈[0,L], δ is the delta function, and $C_{0}$ is the equilibrium initial concentration of CDC-42. $\phi_{a}\left(x\right)$ and $\phi_{c}\left(x\right)$ are perturbation functions, $\epsilon_{p}$ is the strength of the initial external perturbation signal and ϵ is the magnitude of the perturbation.

### 2.5. Minimal Network Analysis

To determine the minimal set of interactions required for maintenance (Section 3.1.1) or generation (Section 3.2.1) of a polarization pattern, we devised a method of minimal network analysis. Starting from the fundamental network containing all interactions (Figure 1C), we first remove individual interactions from the network. This reduced network is simulated and evaluated for the presence of a pattern using initial conditions described above. The total simulation time for pattern evaluation is 16 min for maintenance of polarization and 30 min for generation of polarization. While the numerical solution may not have converged to a stationary solution by the time of evaluation, it is clear whether a pattern has been initiated. We determine which interactions are common to all reduced networks that are capable of maintaining/generating polarization. A network consisting of these common interactions is tested for its capacity to maintain/generate a polarized pattern. If the common interaction network is not capable of maintaining/generating a pattern consistent with polarization, interactions are reintroduced into the model, first individually and then in pairs, and the model is simulated and evaluated for the presence of a polarized pattern. The reduced network(s) with the least number of interactions that is capable of maintaining/generating polarization is retained for further analysis.

### 2.6. eFAST Sensitivity Analysis

Sensitivity analysis is used to determine the influence of model parameters on the dynamics of the network via the use of Extended Fourier Amplitude Sensitivity Test (eFAST) [31,32]. In brief, we define sensitivity functions whose dynamics reveals self-organization and track their changes in response to changes in parameter values. In particular, the sensitivity functions for aPAR, pPAR, and CDC-42, denoted by $F_{S}^{A},F_{S}^{P}$ and $F_{S}^{C}$, respectively, are defined as
(6)$$\begin{array}{c} F_{S}^{A}=\sqrt{\int_{0}^{L}{\left|\frac{\left[A_{m}\right]\left(x,t^{*}\right)-\left[A_{m}\right]\left(x,0\right)}{t^{*}}\right|}^{2}},\\ F_{S}^{P}=\sqrt{\int_{0}^{L}{\left|\frac{\left[P_{m}\right]\left(x,t^{*}\right)-\left[P_{m}\right]\left(x,0\right)}{t^{*}}\right|}^{2}},\\ F_{S}^{C}=\sqrt{\int_{0}^{L}{\left|\frac{\left[C_{m}\right]\left(x,t^{*}\right)-\left[C_{m}\right]\left(x,0\right)}{t^{*}}\right|}^{2}}, \end{array}$$
where $t^{*}$ is the time scale of interest. This time scale is chosen as 16 min for maintenance of polarization, and 30 min for generation of polarization, consistent with experimental observations [33,34]. These sensitivity functions give a quantitative measure of how much the polarity pattern (the model output) has been altered in response to changes in parameter values (the model input). We calculate two sensitivity measures, the first order index ($S_{i}$) and total-effect index ($S_{T_{i}}$). These measures are defined as follows:$$\begin{array}{cc} \; & S_{i}\equiv \frac{\mathrm{Variance}\;\mathrm{of}\;\mathrm{the}\;\mathrm{expected}\;\mathrm{model}\;\mathrm{output}\;y\;\mathrm{with}\;\mathrm{respect}\;\mathrm{to}\;\mathrm{parameter}\;p_{i}}{\mathrm{Total}\;\mathrm{variance}}=\frac{V_{i}}{V_{total}},\\ \; & S_{T_{i}}\equiv \mathrm{Total}\;\mathrm{effect}\;\left(\mathrm{contribution}\right)\;\mathrm{of}\;\mathrm{parameter}\;p_{i}\;\mathrm{to}\;\mathrm{the}\;\mathrm{output}\;\mathrm{variance}=1-\frac{V_{-i}}{V_{total}}, \end{array}$$
where $V_{-i}$ is the effect of any order that does not include the factor *i*. The first order index indicates the influence of parameter $p_{i}$ on the variance of the polarization measure (the model output), independent of interactions with the other parameters. The total-effect index indicates the effect of parameter $p_{i}$ when interactions with the other parameters are included. These two measures give a full quantification of the importance of parameter $p_{i}$ and whether the extent whether this is a direct influence or through interactions with other parameters or both. In this study, we focus on the parameters $q_{3}^{a},\;q_{3}^{c},\;k_{3}^{p},\;k_{3}^{a}$ and $k_{3}^{c}$ which determine the magnitude of the interaction functions, Equation (Equation 2).

A more in depth discussion of the eFAST sensitivity method and its detailed accompanying calculations can be found in Section A.3. The values used for sensitivity analysis are given in Table A2.

## 3. Results

### 3.1. Critical Network Interactions and Parameters for Maintenance of Par Protein Polarization

It has been observed that CDC-42 is required during maintenance but not establishment of polarization in the early *C. elegans* embryo [18,19]. However, it is not clear how CDC-42 is interacting with or regulating the anterior and posterior Par proteins to ensure maintenance of polarization in the early embryo.

#### 3.1.1. Minimal Network for Maintenance of Par Protein Polarization

Using our method of minimal network analysis, we aim to determine the minimal interaction network between anterior Par proteins (aPAR), CDC-42 and posterior Par proteins (pPAR) that may maintain spatial polarization. As shown in Figure 3A, the activation of aPAR by CDC-42 ((a1), $q_{3}^{a}$ interaction) and inhibition of pPAR by aPAR ((a3), $k_{3}^{p}$ interaction) are always present in reduced networks that are able to maintain polarization, suggesting these interactions are critical. The ability of the model to polarize in the absence of the other interactions ((a2), (a4), (a5)) show these interactions are less critical for polarity maintenance.

A minimal network consisting of only these two interactions ($q_{3}^{a}$ and $k_{3}^{p}$, Figure 3C) is not sufficient to maintain polarization. As shown in Figure 3B, adding CDC-42 inhibition by pPAR ((b1), $k_{3}^{c}$ interaction) or aPAR inhibition by pPAR ((b2), $k_{3}^{a}$ interaction) back to the network also cannot maintain polarization. Adding CDC-42 activation by aPAR ((b3), $q_{3}^{c}$ interaction) allows polarity maintenance, although pPAR does not vary much over the domain. Further reinforcing the importance of CDC-42 for maintenance of polarity, we find that polarization of aPAR can be maintained with only two interactions (Figure 3D, $q_{3}^{a}$ and $k_{3}^{p}$ networks) and without mutual inhibition by pPAR. This is consistent with previous results, showing that aPARs intially polarize during the establishment phase, transiently establishing an anterior Par protein domain. However, this polarization can not be maintained, and the aPAR domain gradually creeps back toward the posterior pole, resulting in an eventual loss of polarity [17,18]. Our model reproduces the initial aPAR polarization in the absence of pPAR.

Taken together, the results suggest that the minimal network that is capable of maintaining polarization of aPAR and pPAR involves mutual activation of aPAR and CDC-42 and aPAR-mediated inhibition of pPAR (Figure 3B(b3)).

#### 3.1.2. Critical Parameters for Maintenance of Par Protein Polarization

To determine the effect of the network interactions on maintenance of Par protein polarity with the fundamental network, we performed a sensitivity analysis using the eFAST method [32,35]. See Section A.3 for more details about this approach. We wish to compare the critical interactions found by minimal network analysis with the important parameters indicated by the sensitivity analysis. We evaluated the first order index, $S_{i}$, for the five parameters governing the magnitude of each possible interaction in the network (Figure 3E). We find that the most important parameters as determined by the sensitivity analysis depends on the choice of sensitivity function. When the sensitivity function for aPAR is used as the model output, we find that aPAR activation by CDC-42 ($q_{3}^{a}$) is the most important parameter, followed by CDC-42 activation by aPAR ($q_{3}^{c}$) and pPAR inhibition by aPAR ($k_{3}^{p}$). When the sensitivity function for pPAR is used as the model output, we find that pPAR inhibition by aPAR ($k_{3}^{p}$) is the most important parameter, followed by mutual activation of aPAR and CDC-42 ($q_{3}^{a}$ and $q_{3}^{c}$) as well as CDC-42 inhibition by pPAR ($k_{3}^{c}$). Using the sensitivity function for CDC-42 as the model output, we find that CDC-42 activation by aPAR ($q_{3}^{c}$) is the most important parameter, followed by pPAR inhibition by aPAR ($k_{3}^{p}$), aPAR activation by CDC-42 ($q_{3}^{a}$), and CDC-42 inhibition by pPAR ($k_{3}^{c}$). Combining the most important parameters for each sensitivity function, we find that they correspond to the network interactions in our minimal network ($q_{3}^{a},q_{3}^{c},$ and $k_{3}^{p}$ interactions, Figure 3B(b3)). Since the minimal network was found independently of the sensitivity analysis, this suggests our minimal network contains only the most critical interactions.

We then evaluated the total-effect index, $S_{T_{i}}$. We find that CDC-42 activation by aPAR ($q_{3}^{c}$ interaction) appears as a high index value with respect to both the aPAR and pPAR sensitivity functions, which we did not find with the first order index (see Section A.4 for further details of index value significance and meaning). When considering the aPAR sensitivity function, the mutual activation of aPAR and CDC-42 ($q_{3}^{a}$ and $q_{3}^{c}$ interactions) have a high total index, indicating that these two mutual activation interactions are very important for the maintenance of aPAR polarity (Figure 3D). Similarly, when considering the pPAR sensitivity function, the two high index values correspond to $q_{3}^{c}$ and $k_{3}^{p}$ interactions, suggesting that the activation of CDC-42 by aPAR plays an important role in the maintenance of pPAR polarity along with the inhibition of pPAR by aPAR. When considering the sensitivity function for CDC-42, the total-effect index followed the same trend as the first order index, with CDC-42 activation by aPAR ($k_{3}^{c}$ interaction) being the most important parameter. When considering the total effect of a parameter, this suggests that CDC-42 plays a critical role for maintenance of both aPAR and pPAR polarity. We also found that the magnitude of the sensitivity indices for $S_{i}$ and $S_{T_{i}}$ are substantially different for the pPAR and CDC-42 cases, but not for the aPAR case, suggesting that maintenance of pPAR and CDC-42 polarity are more strongly affected by interactions with the other proteins in the network, but maintenance of aPAR polarity operates largely independently of the other proteins. Together, this indicates that aPAR may play a central role for the entire network during maintenance of polarity, and that maintenance of pPAR polarity may depend directly on maintenance of aPAR polarity.

### 3.2. Critical Network Interactions and Parameters for Generation of Par Protein Polarization

In the previous section, we determined the critical interactions and parameters in a minimal network for the maintenance of Par protein polarity in the early *C. elegans* embryo. Under wild type conditions, establishment of polarization in the early *C. elegans* embryo relies on actin and myosin based advective flow [17,36]. However, polarization can still occur even in the absence of cortical flow [30], although on a longer time scale. Thus, using a similar approach as in the previous section, we aim to determine if the interactions and parameters required for maintenance of polarity are different from those required to generate polarization in the absence of cortical flow.

#### 3.2.1. Minimal Network for Generation of Par Protein Polarization

To determine the minimal network required to generate a polarity pattern, we simulated networks with individual interactions missing to determine which interactions are predicted to be necessary for self-organization. Simulations were given a local stimulus as an initial conditions, as shown in Figure 2B, first panel, and were assessed at 30 min for the presence of polarization. As shown in Figure 4A, the three interactions represented by the parameters $q_{3}^{a},q_{3}^{c},$ and $k_{3}^{p}$ play critical roles in generation of polarization as their absence leads to the absence of pattern generation and these are the same three, interactions required for maintenance of polarization (Figure 3B(b3)). However, the minimal network for maintenance of polarization (Figure 3B(b3)) has not been sufficient to generate a polarized pattern (Figure 4B), and the addition of either the $k_{3}^{a}$ or $k_{3}^{c}$ interaction was also important (Figure 4A(a4,a5)). We have also confirmed that the minimal network for aPAR polarity maintenance ($q_{3}^{a}$ and $q_{3}^{c}$ interactions, Figure 3D) is not sufficient to generate a polarity pattern even in the presence of CDC-42 inhibition by pPAR (Figure 4C). These results suggest that in addition to the minimal network for maintenance phase, aPAR inhibition by pPAR is critical for generating Par protein polarity, and the mutual inhibition between the anterior and posterior Par proteins directly and via CDC-42 is important in generating the pattern.

#### 3.2.2. Critical Parameters for Generation of Par Protein Polarization

We then used sensitivity analysis to determine which parameters are critical for generation of polarization. We used the same sensitivity functions as above (Equation (Equation 6)), and assessed the model simulation at $t^{*}=30$ min. In contrast to our sensitivity analysis results for the maintenance of polarization, we find that CDC-42 activation by aPAR ($q_{3}^{c}$ interaction) is the most important parameter for aPAR, pPAR and CDC-42 with respect to both the first order and total-effect index. Although the interactions represented by the parameters $q_{3}^{a},q_{3}^{c},$ and $k_{3}^{p}$ tended to show higher importance than the other two interactions, $k_{3}^{a}$ and $k_{3}^{c}$, in both first order and total-effect indices, the effect of the $k_{3}^{a}$ and $k_{3}^{c}$ interactions are not negligible in the total-effect index as compared to the first order index for aPAR and CDC-42. This indicates that the $k_{3}^{a}$ and $k_{3}^{c}$ interactions may influence the generation of the polarity pattern, consistent with our minimal network analysis (Figure 4A(a4,a5)).

## 4. Discussion

Through minimal network analysis and eFAST sensitivity analysis, we identified different roles for aPAR, pPAR and CDC-42 in maintaining or generating the spatial pattern associated with polarization. To focus on the biochemical interactions, we focus on the dynamics of polarization in the absence of cortical flow, and neglect interactions between CDC-42 and the Par proteins with mechanical proteins including actin and myosin. Results are summarized in Figure 5 and Table 1. CDC-42 is required in both pattern maintenance and generation: CDC-42 reinforces maintenance of aPAR polarity which in turn directs pPAR polarization as indicated by the high sensitivities of the latter to the up-regulation of CDC-42 and down-regulation of pPAR by aPAR. Thus, the entire system of interactions relies on aPAR polarity to maintain polarization in all variables. However, the minimal network capable of maintaining Par protein polarization is insufficient to generate the polarization pattern, and additional inhibition of aPAR or CDC-42 by pPAR is required. This additional interaction acts to balance the mutual inhibition between aPAR and pPAR, allowing a pattern to be generated. In other words, the posterior Par proteins play a more significant role when generating the polarity pattern as compared to maintaining an established pattern. Interactions between CDC-42 and the anterior Par proteins are critical in both situations (Figure 5 and Table 1). This is consistent with observations suggesting the importance of anterior Par proteins and CDC-42 in generating polarization [37]. This suggests that pPAR, with the support of CDC-42, plays a critical role in generating polarization, though, unlike maintenance, we found all interactions between CDC-42, pPAR and aPAR suggested by previous observation, and thus under consideration, can have some influence on polarization generation.

Remarkably, the minimal network analysis and eFAST both selected the same critical interactions and parameters in polarization maintenance and generation. This independent determination suggests the minimal networks found here represents the key interactions involved in patterning associated with polarization.

In this study, we have made a number of simplifying assumptions to investigate the core interactions between the Rho protein CDC-42 and members of the Par protein family. In future investigations, we will explicitly consider separate members of the Par protein family, each of which have distinct dynamics. For instance, the anterior Par protein PAR-3 appears to compete with CDC-42 to bind to a complex of PAR-6 and aPKC [18], and in the posterior, PAR-2 must be recruited to the membrane before PAR-1 can bind [10]. We will add other members of the Rho protein family, including Rac and Rho, which are known to act in a network with CDC-42 [38]. In this investigation, we have explored protein interactions within the pattern maintenance phase of early embryo development, when advective flow has largely ceased [17,22], and in addition considered the differences in protein interactions required for pattern generation in the absence of cortical flow, as observed in select experimental systems. Future extensions of the model will include the spatial dynamics of advective flow, which may act with the biochemical interaction network to establish polarity. In this study, we have reduced the geometry of the embryo to a 1D domain for computational efficiency to facilitate the exploration of a five dimensional parameter space. However, other studies have studied polarization dynamics of simplified two variable models on fully 3D domains [39] or explicitly included the geometry of the embryo [40] to investigate the spatial orientation of the polarity axis. Noting the computational demands, we leave minimal network and eFAST analysis of higher dimensional parameter spaces in more complex geometries and 3D domains for future work.

In this investigation, we have developed and analysed a mathematical model of the biochemical network that integrates components of two protein families, Par proteins and Rho GTPases, that are known polarity determinants. The minimal network and sensitivity analyses have identified critical interactions in this network, providing testable hypotheses for future experimental work. By resolving the critical role of CDC-42 and highlighting the most important PAR interactions in this network, we have extended our understanding of the signalling network responsible for polarization, and laid the foundation for further investigations into patterning associated with polarization.

## Figures and Tables

**Figure 1 cells-09-02036-f001:**
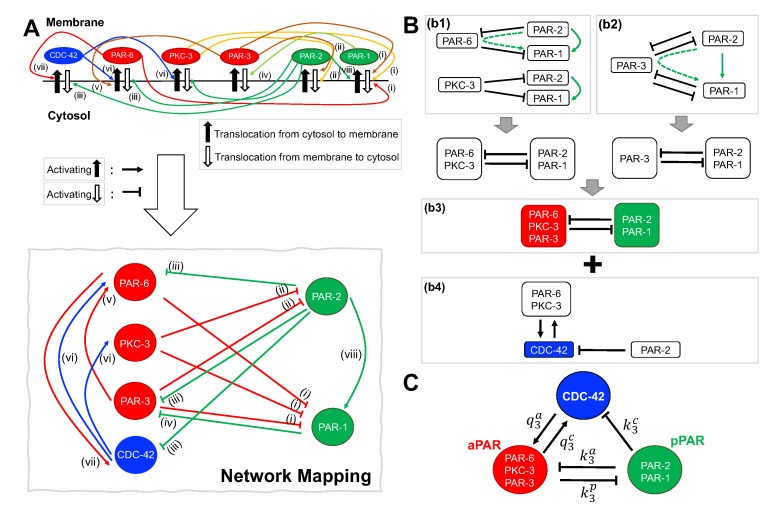
Interaction network of PAR proteins and the Rho GTPase CDC-42. (**A**) Top: Interaction network between PAR proteins and CDC-42 consistent with experimental results. References for interactions (i)–(viii) are given in Section 2.1 of the main text. Bottom: Interaction network used in this study, with cytosol to membrane translocation represented by activation, and membrane to cytosol translocation represented by inhibition. (**B**) Reduction of interaction network in (**A**) into subnetworks. The interaction network is separated into three modules: (b1) shows the interactions between the anterior Par proteins PAR-6 and PKC-3 and the posterior Par proteins PAR-1 and PAR-2, (b2) shows the interactions between the anterior Par protein PAR-3 and the posterior Par proteins PAR-1 and PAR-2, and (b4) shows the interactions between the anterior and posterior Par proteins and the Rho GTPase CDC-42. Note that the upregulation of PAR-6 by PAR-3 is implicitly captured by the inhibitory action of PAR-3 on PAR-2, since the latter in turn downregulates PAR-6. As described in the text, (b1) and (b2) can be reduced to (b3). The green dotted arrows indicates two sequential inhibition interactions (⊤⊥) that are equivalent to an activating interaction (↑, solid green line). (**C**) Network (b3) merged with (b4) is the fundamental network investigated here. Rate constants $q_{3}^{a},q_{3}^{c},k_{3}^{p},k_{3}^{a},$ and $k_{3}^{c}$ appear in the corresponding model Equations (Equation 4).

**Figure 2 cells-09-02036-f002:**
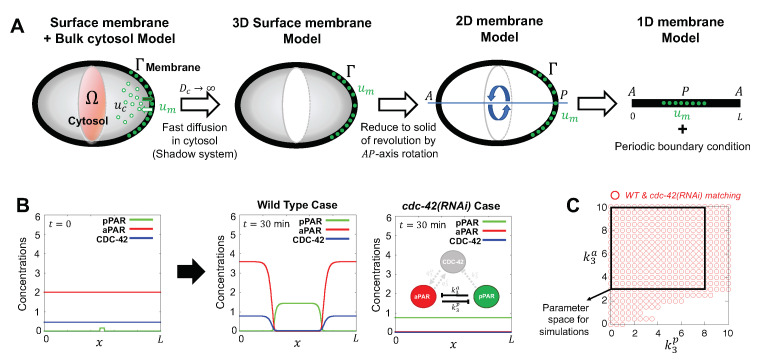
Schematic of model components and geometry. (**A**) Model reduction from three dimensions to one dimension in space. A cross section of the bulk cytosol region is coloured by red, the surface is coloured by gray, with $u_{c},\;u_{m}$ denoting generic cytosolic and membrane densities, respectively. Fast cytosolic diffusion timescales compared to the timescales of interest lead to effectively homogeneous cytosolic concentrations, that can be captured by membrane concentrations via conservation principles, as discussed in Section 2.2. This allows the model to be reduced to membrane equations, with axisymmetry about the major axis of the cell for a simple geometry yielding the final simplification, as detailed in Section A.1. (**B**) Representative simulations for wild type case and *cdc-42(RNAi)* case. Simulations start with initial conditions given by initial conditions, Equation (Equation 5). After 30 min, the solutions shown have reached steady state. Here, and throughout the paper, concentrations are in units of M/L, where M is the characteristic protein number of Table A1, and *L* is the perimeter of the projected membrane in (**A**). (**C**) The range of values for the parameters $k_{3}^{p}$, and $k_{3}^{a}$ that satisfy both wild type and *cdc-42(RNAi)* behaviours which serves as a validation of the model against observations. The red circles indicate the parameter region matching both wild type and *cdc-42(RNAi)* behaviours. The black square indicates the range of $\left(k_{3}^{p},k_{3}^{a}\right)$ used for sensitivity analysis.

**Figure 3 cells-09-02036-f003:**
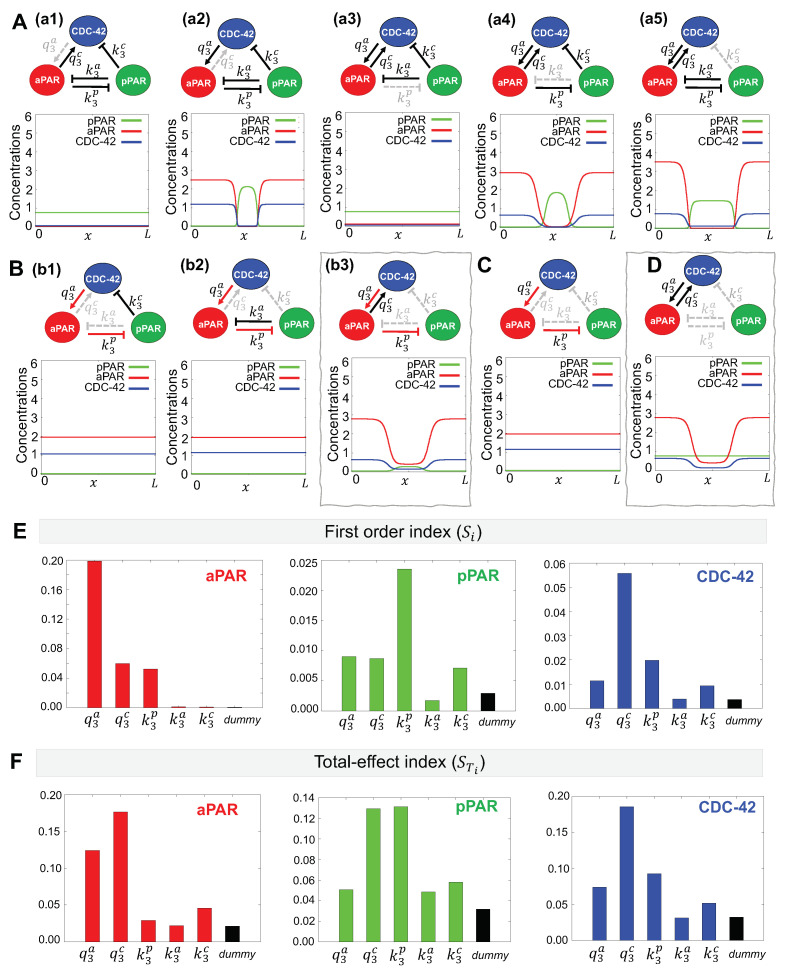
Minimal network needed for maintenance of Par protein polarization. (**A**–**D**) Representative simulations for the effects of network interactions. Dotted gray lines indicate interactions excluded from the network. Simulations are shown at t=16 min, the approximate amount of time the early *C. elegans* embryo spends in the maintenance phase [33]. Initial conditions are given by the corresponding stationary solution (Figure 2B). (**A**) (a1)–(a5) Simulation results of networks omitting a single interaction. Minimal networks that produce the correct spatial pattern and are retained for further analysis are boxed in gray. (**B**) Red arrows indicate interactions identified in (**A**) that are retained in these additional networks. (b1)–(b3) Simulation results of networks omitting two interactions. (**E**,**F**) Variance-based sensitivity analysis results using eFAST. Parameters with higher values have a more significant effect on the network dynamics. (**E**) The first order index (the effect of a parameter on the model dynamics independent of the other parameters) and (**F**) The total effect index (the effect of a parameter including the interactions with other all model parameters) both indicate that CDC-42 interactions with aPAR are critical for producing the correct spatial pattern.

**Figure 4 cells-09-02036-f004:**
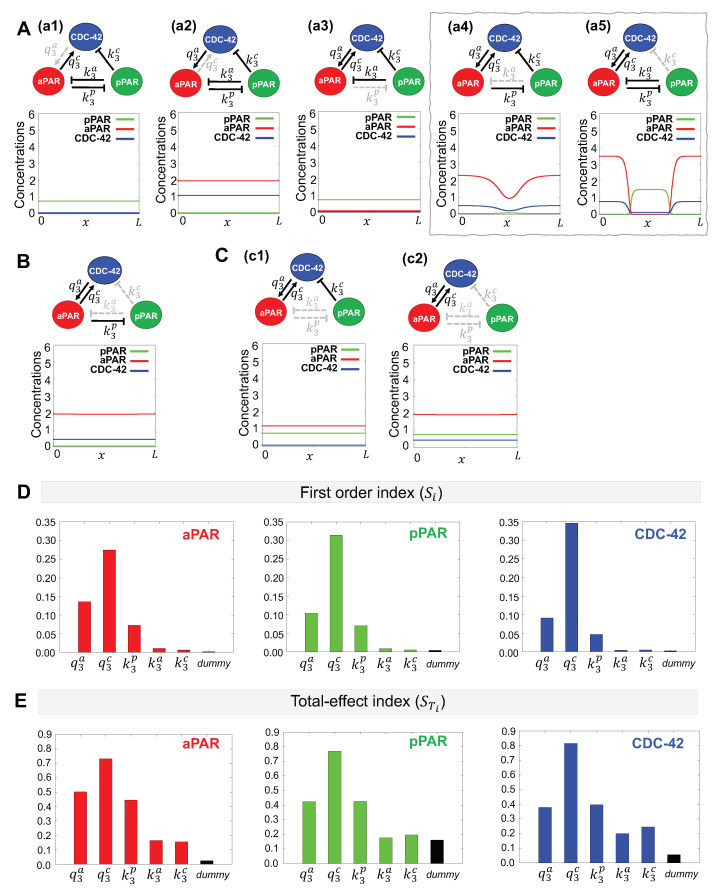
Minimal network needed for generation of Par protein polarization. (**A**–**C**) Representative simulations for the indicated network. Dotted gray lines indicate interactions excluded from the network. The initial condition provides a small local perturbation, as discussed in Section 2.4. Simulation results are shown at t=30, longer than the time scale for polarity emergence reported experimentally [30]. (**A**) (a1)–(a5) Simulation results of networks omitting a single interaction. Networks boxed in gray are minimal networks capable of generating a polarization pattern. (**B**) and (**C**) (c1)–(c2) Simulation results of networks omitting two or more interactions. (**E**–**F**) Variance-based sensitivity analysis results using the eFAST method. Both the first order and total effect index indicate the importance of CDC-42 in producing the correct spatial pattern.

**Figure 5 cells-09-02036-f005:**
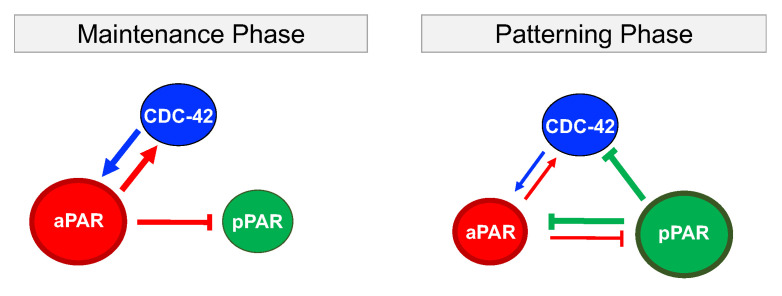
Critical interactions during maintenance and generation of Par protein polarization. A bigger circle indicates a key role for that protein in the indicated phase. In the maintenance phase, aPAR plays a key role in maintaining spatial polarity via interactions with CDC-42. aPAR enforces pPAR polarity through mutual inhibition. In the patterning phase, pPAR is playing the key role by inhibiting aPAR through either direct or CDC-42-mediated inhibition. CDC-42 is indispensable for both the maintenance and emergence of polarization, but does not play a critical role in the dynamics of either phase.

**Table 1 cells-09-02036-t001:** Summary of minimal network and sensitivity analyses results. MNA: Minimal Network Analysis, $S_{i}$: First order index, $S_{T_{i}}$: Total effect index. For MNA, a small checkmark indicates additional interactions required for pattern generation. For sensitivity measures, a/p/c denote the aPAR/pPAR/CDC-42 sensitivity functions, respectively, and a capital letter indicates the parameter with the highest index for that function.

	Maintenance				Generation		
Parameter/Interaction	MNA	$S_{i}$	$S_{T_{i}}$		MNA	$S_{i}$	$S_{T_{i}}$
$q_{3}^{a}:$ CDC-42→aPAR	✓	Apc	a		✓	apc	apc
$q_{3}^{c}:$ aPAR→CDC-42	✓	apC	ApC		✓	APC	APC
$k_{3}^{p}:$ aPAR⊣pPAR	✓	aPc	Pc		✓	apc	apc
$k_{3}^{a}:$ pPAR⊣aPAR			p		✓		ac
$k_{3}^{c}:$ pPAR⊣CDC-42		pc	apc		✓		ac

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
