# Peer review of "CDC-42 Interactions with Par Proteins Are Critical for Proper Patterning in Polarization"

_cells, 2020, doi:10.3390/cells9092036_

Round 1
Reviewer 1 Report
In the manuscript, the authors studied the intracellular localization of conserved polarity effectors, PAR proteins, which is essential for regulating signal transduction pathways in many aspects of embryonic development. They developed a mathematical model of PAR biochemistry, with the aim of revealing a role of CDC-42, a member of RHO family, which has been experimentally demonstrated to play a pivotal role in PAR mediated cell polarity. Basically the manuscript is technically sound, and well organized. However, some aspects needs to be improved before publication.
(1) The authors developed the mathematical model through reducing inter-molecular regulations between PARs and CDC-42 known in literatures. However, some regulatory reactions listed in 2.1 are not shown experimentally, as far as I know.
(1a) In (iii), PAR-2 dependent dissociation of PAR-3 is not direct but mediated through PAR-1 (Hao et al., 2005). Given this, PAR-6, a binding partner of PAR-3 can be removed from the cortex. Does PAR-2 or even PAR-1 contribute to exclusion of PAR-6 directly? Same for PAR-1 dissociation by the action of PAR-3/PKC-3/PAR6 given in (i), PAR-2 dissociation by PAR-3, and so on. Although I know that some of them are not relevant in constructing their model, these should be corrected.
(1b) It is not clear to me which observations lend credence to the assumptions of PAR mediated CDC-42 regulations. Cortical CDC-42 concentration remains unchanged by PAR-2 depletion (Motegi and Sugimoto, 2006). In addition, PAR-6 depletion does not affect cortical CDC-42 density so much (Schonegg and Hyman, 2006). The authors should mention in more detail. It is not sufficient just to cite papers.
(2) Goehring et al. demonstrated that localized distribution of aPAR in anterior cannot be maintained in the absence of posterior PAR, which seems not to be recapitulated in the present study (fig. 3d). This should be argued in the manuscript.
Reviewer 2 Report
In this article, the authors present a mathematical model and detailed minimal network analysis and sensitivity analysis to understand the role of the interactions between Par and Rho-family proteins (namely CDC-42 from the Rho-family and various anterior and posterior PARs) in the generation and maintenance of polarization in the early C. elegans embryo.
The authors formulate a model for the possible interactions between CDC-42, aPARs, and pPARs based on the signaling network elucidated by a large body of experimental work. The model consists of reaction-diffusion partial differential equations (PDE) formulated in the 3D embryo, with bulk-membrane coupling that models how the proteins bind and unbind from the cell membrane. The authors reduce the model to a 1D dimensional model with periodic boundary conditions by exploiting the geometry and timescales in the system. To determine which protein interactions are required for the generation and maintenance of signaling, the authors simulate the model with a small perturbation as an initial condition or an already polarized pattern, respectively, and assess whether the model output matches the wild type protein distribution while interactions between proteins are turned “on” or “off” systematically. This minimal network analysis reveals which sets of interactions are critical for polarization maintenance and generation. The authors next use sensitivity analysis (eFAST), independent of the minimal network analysis, to examine the extent that each of these interactions have in polarization generation and maintenance on their own and with interactions with the other interactions. The sensitivity analysis agrees with the results from minimal network analysis, revealing the critical interactions in both the maintenance and generation of polarity.
The article is clearly organized and is very easy to follow. This is in part due to the very clear consistent figure layout and coloring as well as the detailed, methodical steps taken in the model reduction. The application of the minimal network and sensitivity analysis in the case is particularly appropriate and reveals (beyond typical modeling parameter sweeps for PDE models where only simulations are run over many parameter sets) to what extent these interactions are responsible for the role. The authors conclusions are sound under the assumptions and modeling approach. However, I have some questions and comments that should be addressed before publication.
Major comments:
- The sentence in the abstract “While polarization has been observed in the absence of cortical flow, the identified maintenance mechanisms are also predicted to be insufficient for supporting the generation of polarization in this setting, and additional inhibitory influences of posterior PAR proteins are predicted to play a role” is confusing. Which are “the identified maintenance mechanisms?” From literature or the mechanisms identified by the network and sensitivity analysis?
- It is well-known that cortical flow plays an important role in the polarization processes and that CDC-42 interacts with actin and myosin by generating yet this model does not incorporate cortical flow. The authors justify the omission by focusing on the biochemical interactions and the maintenance phase when flow has largely stopped which is fair. Nonetheless, recent studies (by Dickinson et al., Rodriguez et al., and Wang et al. 2017, previewed in Protein Clustering Shapes Polarity Protein Gradients http://doi.org/10.1016/j.devcel.2017.08.006) do suggest a critical role for aPAR and CDC-42 interactions in the generation of polarity patterns. This is in apparent contrast to the result shown in Figure 5 (right panel) that shows a more critical role for pPARs during patterning. How should this difference be reconciled? The authors may wish to reiterate or expand on the caveat that cortical flow is not considered in the patterning phase.
Minor comments:
- Figure 2C: Do the red circles correspond to the parameter values that match both the cdc-42(RNAi) and wild type behavior? If so, it could be clearer if the label was WT + cdc-42(RNAi) match (or similar) and the white area labeled as not matching.
- Equation 4/5/6: I’m slightly confused about the nondimensional model. In the figures, 0 < x < 1, yet in the equations there is still an L. I thought the domain was non-dimensionalized. Moreover, one integral has 0 L limits while the others are labelled Gamma. Could this non-dimensionalization be clarified?
- What do the red arrows in Figure 3B and C represent (why are they colored red?)?
- Is the simulation shown in Figure 4A (a4) at steady state? The pattern doesn’t appear to have sharp transition layers as expected and observed in other simulations.
- What numerical methods are used to solve the reduced model?
- Please indicate somewhere (maybe in A4 or Figure 3 and 4 or in the main text) whether or not the indices calculated from eFAST are significantly different from zero with respect to the dummy parameter discussed in A4. It is difficult to ascertain the exact values from the bar charts to compare with the values in Table A3.
- I find the motivation for reducing the model to 1D lacking since full 3D simulations of coupled bulk-membrane systems are now common (e.g., A coupled bulk-surface model for cell polarization https://doi.org/10.1016/j.jtbi.2018.09.008 among others). The authors can easily explain that it is only feasible to carry-out the computationally expensive minimal network and eFAST analyses when the PDE simulations are computationally inexpensive (e.g., in 1D). Another recent study does carryout parameter sweeps for PAR polarization in fully 3D finite element simulations of early elegans embryos (Geometric cues stabilise long-axis polarisation of PAR protein patterns in C. elegans https://doi.org/10.1038/s41467-020-14317-w) and may warrant discussion in light of the conclusions here.
- There are some minor spelling and punctuation mistakes. Here is a list of those that I found
- For example, polarization vs polarisation is used inconsistently (pg 1, first two sentences).
- Eqn is not always abbreviated with a period (pg 7, first line), and some equations are missing commas (e.g., (5) and the definitions of the sensitivity measures; first line).
- Artificially is misspelled (‘artifactually’ pg 21, line 384).
- Pg 11, line 211: Does ‘significantly’ refer to a statistic significant or just a difference in magnitude?
- Equation (6) is missing a dx in the integrand.
- There is a period missing after equation 2 on pg 8 line 152.
- (Optional) Is it possible to use the values from the sensitivity analysis be used to quantify the sizes of the shapes used to draw Figure 5? This could lead to a more quantitative depiction of the relative importance of each of the interactions instead of a hand-drawn cartoon.
- (Optional) The authors might consider publishing their code as an archived GitHub repository or similar.
